



# Computation of Self-recruitment in Fish Larvae using Forward- and Backward-in-Time Particle Tracking in a Lagrangian Model (SWIM-v2.0) of the Simulated Circulation of Lake Erie (AEM3D-v1.1.2)
Wei Shi[1], Leon Boegman[1], Josef D. Ackerman[2], Shiliang Shan[1,3], and Yingming Zhao[1,4]
[1]Environmental Fluid Dynamics Laboratory, Department of Civil Engineering, Queen's University, Kingston, ON K7L 3N6,
Canada
[2]Physical Ecology Laboratory, Department of Integrative Biology, University of Guelph, Guelph, ON N1G 2W1, Canada
[3]Department of Physics and Space Science, Royal Military College of Canada, Kingston, ON K7K 7B4, Canada
[4]Aquatic Research and Monitoring Section, Ontario Ministry of Natural Resources and Forestry Lake Erie Fishery Station,
Wheatley, ON N0P 2P0, Canada
*Correspondence to*: Wei Shi (weishi.yz@gmail.com)
**Abstract.** Accurately estimating self-recruitment (SR), the fraction of recruits in a location that originated locally, is critical
for understanding population connectivity. Biophysical models have been typically applied to compute SR by releasing a
certain number of larval particles from each assumed source location and tracking them forward in time. However, various
strategies have been employed for releasing these larval particles: including randomly, consistently, or a number
proportional to the location's area or larval production, which causes ambiguous results. We demonstrate, using theoretical
arguments and numerical simulations from Lake Whitefish (*Coregonus clupeaformis*) larvae in Lake Erie, that SR depends
on larval production at each source location. This dependency suggests that SR may not be computed unambiguously in
these models unless realistic larval production is released from all potential source locations. In contrast, parentage analysis
studies typically computed SR by assessing the fraction of sampled juveniles that originate locally at a settlement location,
instead of identifying larval production at all sources. Therefore, tracking larval particles backward from the settlement
location is proposed as a straightforward approach for computing SR. Our findings demonstrate that SR is independent of the
number of larval recruits at the settlement location, supporting the employment of backtracking models with randomly
released larval particles. In this way, considerable effort and resources, that would otherwise be spent on identifying all
potential sources and their larval output, in forward tracking can be saved. We believe this result will have important
implications for studies on larval dispersal and recruitment in aquatic systems.
## 1 Introduction
Most marine species have a pelagic larval phase, during which larvae are transported by currents away from a source
population and subsequently recruited into a receiving population, thereby regulating population connectivity (Cowen and





Sponaugle, 2009; Arevalo et al., 2023; Wood et al., 2014). Since Cowen et al. (2000) stated that marine populations may not
be as open as previously thought, there has been accumulating evidence that the probability of dispersal declines rapidly with
distance (Almany et al., 2013; Buston and D'aloia, 2013). Furthermore, high values of self-recruitment (SR) and local
retention (LR) may be common in many fish populations (Cowen et al., 2000; James et al., 2002; Cowen et al., 2006;
Almany et al., 2007; Hamilton et al., 2008; Hogan et al., 2012). This indicates that management decisions, based on open
population models, might overestimate larval exchange, potentially leading to mismanagement of both local and downstream
populations (Cowen et al., 2000; Nanninga et al., 2015). Therefore, measuring SR and LR is essential for quantifying
localized recruitment, assessing the self-replenishment and persistence of populations, and designing effective fisheries
management plans (D'aloia et al., 2013; Burgess et al., 2014; Lett et al., 2015).
SR is defined as the fraction of all recruits at a location that originated locally (Botsford et al., 2009); it reflects
regional replenishment and the openness to recruitment from other locations (Burgess et al., 2014; Lett et al., 2015). LR
indicates the self-persistence of a population, in the absence of external propagule inputs (Burgess et al., 2014). LR has been
defined as the ratio of locally produced setters to total local larval release (Botsford et al., 2009) or as the ratio of locally
produced settlers to the total number of locally released larvae, that successfully settle in suitable nursery locations and
survive (Hogan et al., 2012). Here, we define the latter, which includes only successfully settled larvae, as local retention,
while the former, which encompasses both successful and unsuccessful settlers (those settling in unsuitable nursery
locations), is termed "theoretical" local retention (TLR), as described by Shi et al. (2024). The three metrics (LR, TLR, and
SR) share the same numerator, representing the number of local settlers, but differ in their denominators.
Parentage analysis and/or larval tagging have been widely used to estimate LR, TLR and SR (Jones et al., 1999;
Pinsky et al., 2012; D'aloia et al., 2013; Lett et al., 2015; Planes et al., 2009). By assigning sampled juveniles to their parents
according to DNA relationships, researchers can identify their source locations and quantify the number of settlers
originating from each source location. However, the total number of eggs/larvae produced at a source location $i$, $N_i'$, and their
survival rates remain poorly understood, making it challenging to empirically assess LR and TLR (Lett et al., 2015).
Consequently, biophysical models, typically coupled with forward-in-time Lagrangian particle tracking models (herein
referred to as forward tracking models), have been widely applied to study larval dispersal and compute TLR and LR
(Chaput et al., 2022; Saint-Amand et al., 2023; Sato et al., 2023; Gurdek-Bas et al., 2022). By assuming that $N_i$ larvae are
produced at location $i$, i.e., assuming $N_i' = N_i$, releasing $N_i$ larval particles from $i$ and tracking them forward in time, TLR
can be computed as the ratio of the number of larvae that settle at location $i$ to $N_i$. LR can also be computed by excluding the
larvae that settle in unsuitable nursery locations from $N_i$ (Gurdek-Bas et al., 2022). It is worth noting that both LR and TLR
at location $i$ are independent of $N_i'$, making it effective to release a random number of larval particles $N_i$ from the location, as
will be demonstrated in this research.
Typically, SR has also been computed using biophysical models (Paris et al., 2005; Hiddink et al., 2013; Dubois et
al., 2016; Klein et al., 2016; Faillettaz et al., 2018; Lequeux et al., 2018; Meerhoff et al., 2018; Hidalgo et al., 2019;
Wolanski et al., 2021; Saint-Amand et al., 2023; Michie et al., 2024; Nadal et al., 2024; Corrochano-Fraile et al., 2022; Sato





et al., 2023). In this case, a certain number of larval particles are released from each assumed source location and tracked forward in time. SR at the settlement location $j$ is then computed as the number of larvae both released from and settled at $j$, divided by the total number of larvae that settled at $j$. Notably, the denominator is related to the larval production from all source locations, which may transport larvae to $j$. In contrast, the numerator, representing the number of settlers originating from $j$ itself, varies only with the larval production at $j$. Changes in larval production at any of the source locations can thus potentially alter the total number of settlers at $j$, resulting in variation in SR. However, simply identifying all potential source locations poses a challenge and it remains even less understood how many larval particles should be released from each source location.

At least four distinct strategies have been employed for releasing larval particles. For example, Hiddink et al. (2013), Dubois et al. (2016), and Faillettaz et al. (2018) assumed that larval production was consistent across all source locations, where each location was assumed to produce 500 larvae (Dubois et al., 2016), 1500 larvae (Faillettaz et al., 2018), or 10,000 larvae (Hiddink et al., 2013), respectively. In this strategy, the SR at a location of interest was computed independently of the larval production at each source location. Sato et al. (2023) assumed that 900 larvae were produced from each of the 84 source locations at Puerto Galera (PG) in the Verde Island Passage. The 84 source locations were randomly divided into three regions, with 5 locations at PG, 45 locations east of PG, and 34 locations west of PG. Therefore, 4500 larval particles were released from PG, 40500 from the east and 30600 from the west, resulting in 17.9, 4.4, and 71.9 particles settled at PG, respectively, giving a value of SR at region PG as 17.9 / (17.9 + 4.4 + 71.9) = 0.19. However, if PG was divided into more locations, the number of local settlers at PG (i.e., 17.9) may be increased, altering the value of SR. D'agostini et al. (2015) assumed that larger locations produced proportionally more larvae than smaller ones, and Saint-Amand et al. (2023) assumed a constant density of 500 larvae/km$^2$ and a minimum release of 100 larvae for the smallest location. The resultant SR at each location, therefore, depended on the location area. Nolasco et al. (2022) assumed larval production at each location was proportional to the product of the adult abundance score by the spawning intensity score. From these examples, it remains uncertain whether larval production at each location is constant or proportional to the area of the location. Accurately releasing the number of larvae produced at each location may yield a more precise estimation of SR; however, the realistic number of larval productions at each location remains a challenge to observe. Additionally, there may be unknown source locations contributing to unexpected recruitment that is not accounted for in these simulations, causing potentially misleading estimates of SR.

Conversely, Shi et al. (2024) used backward-in-time Lagrangian particle tracking models (SWIM-V2.0, herein referred to as backtracking models) to estimate larval hatching locations of Lake Whitefish (*Coregonus clupeaformis*) and proposed that backtracking models may be more efficient in computing SR. Backtracking models, release larval particles from larval sampling locations and track them in reverse time, providing a straightforward approach to modeling recruitment that has been widely applied to study the spawning/hatching locations of fish larvae (Christensen et al., 2007; Thygesen, 2011; Bauer et al., 2014; Gargano et al., 2022; Rowe et al., 2022; Chaput et al., 2023). In this case, the denominator of SR, the total number of settlers at location $j$, $M_j^{'}$ (which is unknown as well), is assumed to be $M_j$, indicating that $M_j$ larval





particles will be released from *j* and tracked backward in time. It is no longer necessary to identify all potential source
locations and their corresponding larval production for estimating the denominator. The SR at *j* is independent of the real
number of recruits at *j*, making it effective to release a random number ( $M_j$ ) of larval particles from *j*, as will be
demonstrated.
In this research, we show theoretically that in forward tracking simulations, LR and TLR are independent of the
larval production from the source location, while SR is not. Moreover, using forward tracking models to compute SR can
yield ambiguous results. This assertion brings into question the numerous estimates of SR from studies that have employed
different strategies for releasing larval particles from each source location within forward tracking models. Additionally, we
compute SR using backtracking models and show that this SR remains independent of the number of recruits at the
settlement location. We validate these assertions by applying both forward and backtracking models to compute LR and SR
associated with observations of Lake Whitefish (*Coregonus clupeaformis*) larvae sampled in Lake Erie. Our findings are
applicable to both freshwater and marine species that undergo a pelagic larval phase.

## 2 Theoretical Development

### 2.1 Self-recruitment from forward tracking models

Suppose that there are a set of *n* locations associated with larval hatching and larval settling or recruitment (locations 1, 2, …,
*i, j, …, n*). Here, $N_i^{'}$ represents the realistic number of eggs spawned or newly hatched larvae at location *i*. The larvae become
pelagic upon hatching and undergo a dispersal process, being transported away from the source location by water currents.
The dispersal rate from patch *i* to patch *j*, denoted as $D_{ij}$, is defined as the proportion of larvae released from location *i* that
settle at location *j*. The theoretical local retention (TLR) at location *i*, commonly used in forward tracking simulations (Saint-
Amand et al., 2023; Sato et al., 2023), is defined as follows, as per Shi et al. (2024):
$$\text{TLR}_i = \frac{D_{ii} N_i^{'}}{N_i^{'}} = D_{ii} \text{ ,} \qquad\qquad (1)$$
At the end of larval dispersal, some larvae settle at suitable nursery locations, while some settle in unsuitable ones;
the latter are referred to as 'unsuccessful' settlers (Almany et al., 2017). By excluding the unsuccessful settlers from the
denominator of TLR, we obtain local retention (LR), which is also known as relative local retention (Hogan et al., 2012; Lett
et al., 2015):
$$\text{LR}_i = \frac{D_{ii} N_i^{'}}{\sum_{j=1}^{n} D_{ij} N_i^{'}} = \frac{D_{ii}}{\sum_{j=1}^{n} D_{ij}} \text{ ,} \qquad\qquad (2)$$
Self-recruitment (SR), the ratio of local larval recruitment to all the recruitment at the settlement location, at
location *i* is (Botsford et al., 2009; Lett et al., 2015; Almany et al., 2017):





$\text{SR}_i = \frac{D_{ii}N_i^{'}}{\sum_{j=1}^{n} D_{ji}N_j^{'}},$     (3)
Assuming consistent larval production across all locations, i.e., $N_1^{'} = ... = N_i^{'} = N_j^{'} = ... = N_n^{'}$, the SR can be expressed as
follows:
$\text{SR}_i = \frac{D_{ii}}{\sum_{j=1}^{n} D_{ji}},$     (4)
Here, both TLR and LR are theoretically independent of $N_i^{'}$. When performing the forward tracking simulations,
releasing a random number of larval particles, $N_i$, from the location $i$, is demonstrated as an effective approach to compute
the unambiguous values of TLR and LR. In practice, a sufficiently large number of particles is needed for the dispersal rate
($D$) to converge and accurately reflect the underlying processes. With more particles, the estimate of $D$ stabilizes as it better
captures the full distribution of trajectories. TLR provides a minimum value of LR as stated by Shi et al. (2024), as the
denominator contains both successfully and unsuccessfully settlers. However, SR is shown to be dependent on larval
production at each source location ($N_1^{'}, N_2^{'}, ..., N_i^{'}, N_j^{'}, ..., N_n^{'}$), demonstrating that any strategies for larval particle release
other than releasing the realistic larval production from each source location in forward tracking may not unambiguously
compute SR. Though assuming consistent larval production across all locations can make SR independent of larval
production (Eq. 4), this assumption can lead to significant discrepancies in the estimated value of SR compared to using
realistic value of larval production (Eq. 3), particularly when there are many source locations.
**2.2 Self-recruitment from backward tracking models**
An alternate approach is to release larval particles from the locations where larvae are recruited into the population
(settlement locations) and track them backward-in-time. Suppose that there are a set of $m$ locations associated with larval
hatching and recruitment respectively (locations 1, 2, ..., $i$, $j$, ..., $m$). $M_i^{'}$ larvae are recruited into settlement location $i$. The
recruitment rate $R_{ij}^{'}$ is defined as the proportion of larvae recruited into the settlement location $i$ that originated from the
source location $j$. SR at location $i$ can be written as:
$\text{SR}_i = \frac{R_{ii}^{'}M_i^{'}}{M_i^{'}} = R_{ii}^{'},$     (5)
SR at location $i$ is theoretically equivalent to the recruitment rate at that location, independent of the number of
recruits at $i$. This is reasonable; for instance, in parentage analysis studies, sampling all recruits at a location in the field is
often challenging, SR is thus estimated as the number of sampled juveniles assigned to originate locally based on DNA
relationships, divided by the total number of sampled juveniles at that location (D'aloia et al., 2013; Almany et al., 2017).
This computation of SR, based on sampled recruits, can be used to reflect the overall recruitment dynamics, further
supporting the notion that SR is independent of the number of recruits at the location of interest.





In backtracking simulations, a random number of larval particles $M_i$ are typically released from location $i$. It is
important to note that by the end of the simulation, some particles will have settled in suitable hatching or source locations,
while some will have settled in unsuitable locations, leading to what is termed 'unreal' recruitment (Shi et al., 2024). The
settlement rate $R_{ij}$, is the ratio of the number of larval particles that settle at location $j$ and originate from location $i$, divided
by the total number of larval particles released from location $i$. The difference between $R_{ij}$ and $R'_{ij}$ lies in the denominator,
which corresponds to $M_i$ and $M'_i$, respectively. Substituting $R_{ij}$ and $M_i$ for $R'_{ij}$ and $M'_i$ in Eq. 5 does not yield SR, but rather
the theoretical self-recruitment (TSR), which is the minimum value of SR, as its denominator contains unreal recruits (Shi et
al., 2024):
$$\text{TSR}_i = \frac{R_{ii}M_i}{M_i} = R_{ii} \ , \tag{6}$$
The SR at location $i$, in backtracking simulations, is obtained by excluding the unreal recruits from the denominator of TSR,
expressed as:
$$\text{SR}_i = \frac{R_{ii}M_i}{\sum_{j=1}^{n} R_{ij}M_i} = \frac{R_{ii}}{\sum_{j=1}^{n} R_{ij}} \ , \tag{7}$$
Unlike Eq. (3), SR at location $i$, computed through backtracking models, is theoretically independent of the number
of larval particles released from the location ($M_i$). This independence arises because the settlement rate remains constant
regardless of $M_i$, as will be demonstrated later with numerical data. Consequently, it is effective to release a random
number of particles from the settlement location of interest.
The LR at location $i$ in backtracking simulations is:
$$\text{LR}_i = \frac{R_{ii}M_i}{\sum_{j=1}^{n} R_{ji}M_j} \ , \tag{8}$$
The LR, estimated by backtracking, is dependent on $M_i$, unlike for forward tracking (Eq. 2). Moreover, backtracking cannot
be used to compute TLR as there is no unsuccessful dispersal in the simulation, and similarly, forward tracking cannot
compute TSR as there is no unreal recruitment.

**2.3 Number of larvae produced and recruited at each location**

The number of larvae produced at each location $N'_i$ may be computed for use in forward tracking simulations to obtain
unbiased estimates of SR, based on the number of recruits to location $i$, $M'_i$.
If $M_i$ particles are released from location $i$ in backtracking simulations, $M_i \cdot \sum_{j=1}^{n} R_{ij}$ particles are real recruits, as
they settle in suitable hatching locations; correspondingly, $M_i \cdot (1 - \sum_{j=1}^{n} R_{ij})$ are unreal recruits. Therefore, if $M'_i$ recruits
are sampled at a location $i$, the number of real recruits $M_i \cdot \sum_{j=1}^{n} R_{ij}$ must equal $M'_i$, and $M_i = M'_i / \sum_{j=1}^{n} R_{ij}$ particles should be





released at the location for backtracking. The number of recruits, at location $i$, that originates from location $j$ can thus be obtained as:

$$N_{ji}^{'} = R_{ij} \cdot M_{i}^{'} \Big/ \sum_{j=1}^{n} R_{ij} \,, \tag{9}$$

From the dispersal rate $D_{ji}$, the number of larvae produced at location $j$, is:

$$N_{j}^{'} = \frac{N_{ji}^{'}}{D_{ji}} = M_{i}^{'} \cdot \frac{R_{ij}}{D_{ji}} \cdot \frac{1}{\sum_{j=1}^{n} R_{ij}} \,, \tag{10}$$

Interestingly, if the number of recruits at another location $a$ is $M_{a}^{'}$, then the number of larvae produced at location $j$ can also be written as:

$$N_{j}^{'} = M_{a}^{'} \cdot \frac{R_{aj}}{D_{ja}} \cdot \frac{1}{\sum_{j=1}^{n} R_{aj}} \,, \tag{11}$$

Combining Eq. (10) and (11), we can obtain the number of recruits at location $a$ as:

$$M_{a}^{'} = M_{i}^{'} \cdot \frac{R_{ij}}{D_{ji}} \cdot \frac{D_{ja}}{R_{aj}} \cdot \sum_{j=1}^{n} \frac{R_{aj}}{R_{ij}} \,, \tag{12}$$

Both $N_{j}^{'}$ and $M_{a}^{'}$ are undefined when the dispersal, recruitment or settlement rates become zero.

## 3 Putting Theory into Practice: Application to Lake Erie

### 3.1 Study area

Shi et al. (2024) identified the Lake Whitefish larval hatching locations in Lake Erie from backtracking simulations. The locations were primarily distributed along the western and southern flanks of the western basin. Considering that Lake Whitefish eggs incubate on hard substrates (Amidon et al., 2021), we selected four regions with hard substrates along the western and southern flanks of the western basin as potential hatching locations (Fig. 1). These were the release regions for larval particles in our forward tracking simulations. We refer to these locations as the Detroit River Mouth (region A), Western Shoreline (region B), Midlake Reefs (region C), and Bass Islands (region D). The Midlake Reef and Bass Island regions were also selected as settlement regions, where we released larval particles for backtracking simulations.



Figure 1: (a) Map showing the three basins of Lake Erie: western basin (WB), central basin (CB), and eastern basin (EB). The lake bathymetry was obtained from https://www.ngdc.noaa.gov/mgg/greatlakes/erie.html. (b) Substrate distributions in the western basin from side-scan sonar transects (Haltuch et al., 2000). (c) Color coded Lake Whitefish (*Coregonus clupeaformis*) hatching locations (Detroit River Mouth in red, region A; Western Shoreline in green, region B; Mid-Lake Reefs in yellow, region C; Bass Islands in black, region D), where larval particles were released at the centre of each 500 m × 500 m AEM3D grid (black cross-hatching).

## 3.2 The hydrodynamic model

Larval particles were transported using output from an application of the hydrostatic 3D Reynolds-averaged Navier-Stokes equation model, the Aquatic Ecosystem Model (AEM3D) (www.hydronumerics.com.au). The model simulated the water temperature and currents in Lake Erie during a continuous 2017-2019 hindcast run, using a 500 m × 500 m horizontal grid with 45 vertical layers (Shi et al., 2024; Lin et al., 2022). There was fine resolution (0.5 m) through the surface layer, metalimnion and bottom of the central basin, and coarser resolution layers (5 m) through the hypolimnion of the deeper





eastern basin. The model was forced with surface meteorological data (wind speed and direction, air temperature, relative
humidity and long- and short-wave solar radiation) from four weather stations, had five inflows (Detroit, Maumee, Sandusky,
Cuyahoga and Grand rivers) and the Niagara River outflow. Model calibration and validation were described in the
supplemental material (tables S2-S4) of Shi et al. (2024).
The AEM3D model, and its non-parallel predecessor the Estuary and Lake Computer Model (ELCOM), have been
applied to Lake Erie to backtrack the present Lake Whitefish larval observations and determine hatching locations (Shi et al.,
2024); to hindcast the thermal structure (León et al., 2005), internal wave dynamics (Valipour et al., 2015), surface wave /
sediment transport (Lin et al., 2021), nutrient and chlorophyll-a distributions (Leon et al., 2011), seasonal succession of
phytoplankton groups (Wang et al., 2024); and to forecast storm surge and upwelling/downwelling events (Lin et al., 2022).
**3.3 The Lagrangian particle tracking model**
We used a Matlab®-based Lagrangian particle tracking model (SWIM-v2.0) to study SR. An earlier version of this model
was applied forward-in-time to track silver eel (*Anguilla rostrata* and *Anguilla anguilla*) migration (Béguer-Pon et al., 2016)
and backward-in-time to determine the larval Lake Whitefish hatching locations used in this study (Shi et al., 2024). Diel
vertical migration and active swimming behavior were not considered (Di Stefano et al., 2022; Rowe et al., 2022; Suca et al.,
228 2022).

A horizontal turbulent diffusivity $K_h$ = 0.1 m$^2$s$^{-1}$ and timestep $dt_p$ = 600 s were used in both forward and backward
tracking simulations and larval particles were released at a 3-m water depth and were removed if they encountered the lake
boundary (Shi et al., 2024).  In the forward tracking simulations, particles were released daily at 12:00-noon between 21
March and 8 May 2018 (in four regions; Table A1, regions A, B, C and D) and were tracked for 12 days. In the backtracking
simulations, particles were released daily at 12:00-noon between 2 April and 20 May 2018 (in two regions for 12 days; Table
A1, regions C and D). Each release region was divided into 500 m × 500 m AEM3D grid cells, and particles were released at
the centres of these cells.
**3.4 Nomenclature and data analysis**
In forward tracking, the number of larval particles released from location $i$ was $N_i$ and the number of particles released from
location $i$ that settled at location $j$ was $F_{ij}$. For example, if $N_A$ particles were released from region A; $F_{AC}$ represents the
number of particles that settled in region C that were released from region A. In backtracking, the number of particles
released from location $i$ was $M_i$ and the number of particles released from location $i$ that settled at location $j$ was $B_{ij}$. For
example, if $M_C$ particles were released from region C; $B_{CA}$ represents the number of particles that settled in location A that
were released from region C.
The dispersal rate $D_{ij}$ and settlement rate $R_{ij}$ are given by:
$D_{ij} = F_{ij}/N_i$ ,                                                                                               (13)





$$R_{ij} = B_{ij}/M_i \, ,\tag{14}$$
When computed from forward and backward simulations, local retention and self-recruitment are written as LR_F, LR_B,
SR_F and SR_B, respectively. From Eqs. (11) and (14), the number of larvae produced at each location $N_i'$ is:

$$N_i' = B_{ji}/D_{ij} \, ,\tag{15}$$

250       For example, $N_A'$ is the number of larvae produced from region A. To estimate $N_i'$, from Eq. (11), requires both

backward and forward tracking simulations, and the number of recruits to the location. Here, we only backtracked particles
from region C and D; therefore, $N_i'$ can be computed as $B_{Ci}/D_{iC}$ or $B_{Di}/D_{iD}$, which are referred to as $N_{i\_C}'$ and $N_{i\_D}'$,
respectively. Dividing $N_{A\_C}'$ by the number of AEM3D grid cells in region A (or the total area of the 500 m × 500 m grid
cells) gives the density $d_{A\_C}'$.

## 255 4 Results

As examples of forward and backward trajectories, and to illustrate the validity of the tracking simulations, we show
backtracked larval particles from the Midlake Reefs (region C, Fig. 2a) and forward tracked particles from the Western
Shoreline (region B, Fig. 2b). When particles were released from region C and tracked backward for a period of 12 days,
they mostly settled along the southern and western franks of the western basin (yellow dots in Fig. 2a), consistent with the
settlement distributions in Shi et al. (2024). Most particles were backtracked to regions westward of the release locations
with few travelling to regions eastward, following the predominant west-to-east flow patterns of water movement in the lake
(Beletsky et al., 2013).

263       When particles were released from region B and tracked forward for a period of 12 days, they were mostly

transported to the east of the release locations (green dots in Fig. 2b), also consistent with the flow patterns moving particles
from west to east in the lake. However, some of the particles were transported southeast and northeast of the release
locations, which seems to be counter-intuitive but is not unreasonable given the complex topography and variability in the
wind direction in the region. Adding the northeast release region to the backtracking simulations would, therefore, lead to
backward trajectories to region B.

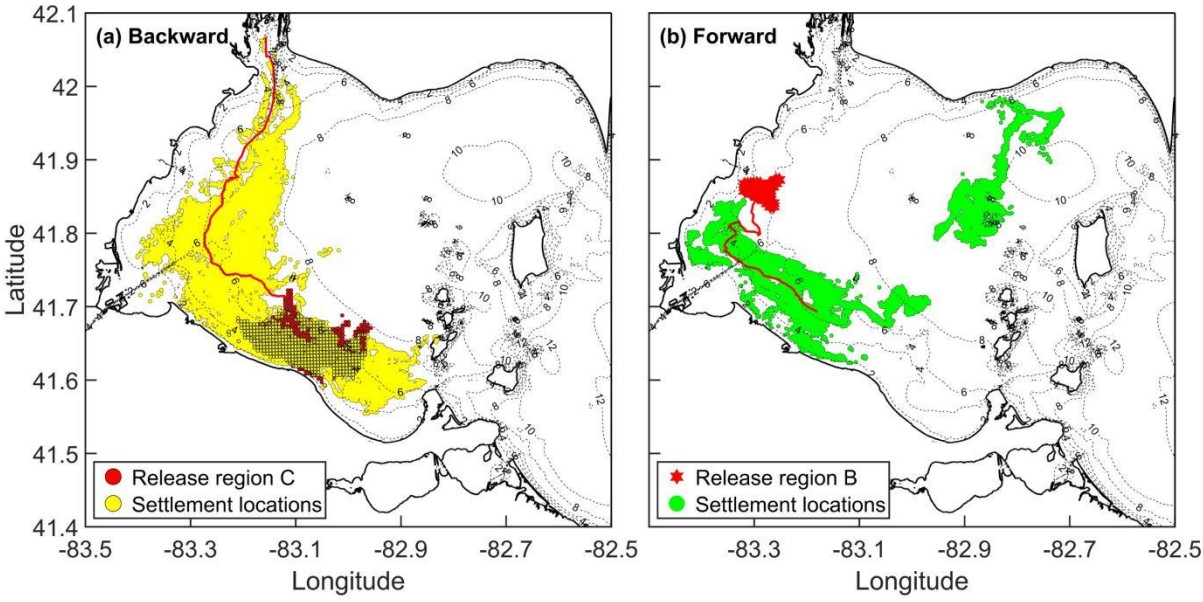

**Figure 2: The settlement location distributions when (a) releasing particles from the red circles and tracking them backward, (b) releasing particles from the red stars and tracking them forward. Yellow circles indicate the settlement locations in backtracking and green circles indicate the settlement locations in forward tracking. Red lines show particle trajectories.**

In the forward tracking simulations, the number of settled larval particles ($F_{ij}$) varied with the number of particles released ($N_i$) and the dispersal rate ($D_{ij}$) was independent of $N_i$ (Table 1). Both TLR and LR had negligible variation with $N_i$, for example TLR_F at region C equaled to $D_{CC}$ as ~ 0.13 and TLR_F at region D equaled to $D_{DD}$ as ~ 0.024. Based on Eq. 3, LR_F equaled to 0 for region A and region B, ~ 0.755 for region C, and ~ 0.99 for region D.

The individual SR_F values are not given in Table 1, because there were 81 different SRs obtained at region C through changing $N_i$, consistent with Eq. (3), ranging from 0.22 to 0.95. For example, when 151200 particles were released from A, 140000 particles from B, 19460 from C and 148400 from D, the SR_F at region C was $2537/(4668+4195+2537+37)=0.22$; whereas when 16800 particles, 17500 particles, 155680 particles, and 18550 particles were released from region A, B, C, and D respectively, SR_F at region C was $20758/(527+501+20758+2)=0.95$. In other words, releasing more particles from region C and fewer particles from the other regions increased the SR at region C, as $D_{CC}$ was much larger than $D_{AC}$, $D_{BC}$ and $D_{DC}$. Indeed, SR can approach 1 or 0 through adjustment of $N_i$. The true value of SR can only be obtained if the actual number of larvae produced at each location is released (i.e., $N_i = N_i'$); however, $N_i'$ remains unknown.

If all four regions released the same number of particles, SR_F would be only a function of $D_{ij}$, and would be independent of $N_i$. For example, the SR_F for region C was $D_{CC}/D_{AC}+D_{BC}+D_{CC}+D_{DC}= 0.68$ and the SR_F for region D was $D_{DD}/D_{AD}+D_{BD}+D_{CD}+D_{DD}= 0.32$.



289          In the backtracking simulations, the number of settled larval particles $B_{ij}$ varied with $M_i$. The settlement rate $R_{ij}$ and

SR_B had negligible variation with $M_i$ (Table 2); showing the SR calculated from backtracking to be independent of $M_i$, as
indicated by Eq. (7). The correct SR values from backtracking may be compared to the erroneous ones from forward
tracking, which had assumed $N_i$ to be the same for all sources. SR_B at region C was larger than SR_F (0.97 vs. 0.68) and
SR_B at region D was smaller (0.18 vs. 0.32) because of the variation in the number of particles released from each location
$N_i$. If the recruits to region C, $M'_C$, was equal to $M_C \cdot (R_{CA} + R_{CB} + R_{CC} + R_{CD})$, then the number of particles that should
have been released in forward tracking from region C should be ~ 2 times greater than those from region D, ~ 10 times
greater than those from region B, and ~ 50 times greater than those from region A (see $N'_{A\_C}$, $N'_{B\_C}$, $N'_{C\_C}$, and $N'_{D\_C}$ in Table
3); assuming the same $N'_i$ increased the denominator of SR_F at region C, and decreased the denominator of SR_F at region
D. When scaled by the number of cells in each region, the larval density from region C was roughly half that from region D,
~4 times that from region B, and ~10 times that from region A (see $d'_{A\_C}$, $d'_{B\_C}$, $d'_{C\_C}$, and $d'_{D\_C}$ in Table 3).

**Table 1.** Number of particles released from the four regions $N_i$, settled to four regions $F_{ij}$, and the dispersal rate $D_{ij}$ in the
forward tracking simulations. The regions are the Detroit River Mouth (region A), Western Shoreline (region B), Midlake
Reefs (region C), and Bass Islands (region D).

| $N_A$ | $F_{AA}$ | $F_{AB}$ | $F_{AC}$ | $F_{AD}$ | $D_{AA}$ | $D_{AB}$ | $D_{AC}$ | $D_{AD}$ | LR_$F_A$ |
|---|---|---|---|---|---|---|---|---|---|
| 16800 | 0 | 0 | 527 | 140 | 0 | 0 | 0.0313 | 0.0083 | 0 |
| 84000 | 0 | 0 | 2587 | 665 | 0 | 0 | 0.0308 | 0.0079 | 0 |
| 151200 | 0 | 0 | 4668 | 1189 | 0 | 0 | 0.0309 | 0.0079 | 0 |
| $N_B$ | $F_{BA}$ | $F_{BB}$ | $F_{BC}$ | $F_{BD}$ | $D_{BA}$ | $D_{BB}$ | $D_{BC}$ | $D_{BD}$ | LR_$F_B$ |
| 17500 | 0 | 0 | 501 | 0 | 0 | 0 | 0.029 | 0 | 0 |
| 70000 | 0 | 0 | 2097 | 0 | 0 | 0 | 0.030 | 0 | 0 |
| 140000 | 0 | 0 | 4195 | 0 | 0 | 0 | 0.030 | 0 | 0 |
| $N_C$ | $F_{CA}$ | $F_{CB}$ | $F_{CC}$ | $F_{CD}$ | $D_{CA}$ | $D_{CB}$ | $D_{CC}$ | $D_{CD}$ | LR_$F_C$ |
| 19460 | 0 | 0 | 2537 | 843 | 0 | 0 | 0.130 | 0.0433 | 0.751 |
| 77840 | 0 | 0 | 10440 | 3327 | 0 | 0 | 0.134 | 0.0427 | 0.758 |
| 155680 | 0 | 0 | 20758 | 6739 | 0 | 0 | 0.133 | 0.0433 | 0.755 |
| $N_D$ | $F_{DA}$ | $F_{DB}$ | $F_{DC}$ | $F_{DD}$ | $D_{DA}$ | $D_{DB}$ | $D_{DC}$ | $D_{DD}$ | LR_$F_D$ |
| 18550 | 0 | 0 | 2 | 435 | 0 | 0 | 0.0001 | 0.0235 | 0.995 |
| 74200 | 0 | 0 | 19 | 1800 | 0 | 0 | 0.0002 | 0.0242 | 0.990 |
| 148400 | 0 | 0 | 37 | 3648 | 0 | 0 | 0.0002 | 0.0246 | 0.990 |



**Table 2.** Number of particles released from two regions $M_i$ and settled in four regions $B_{ij}$, settlement rate $R_{ij}$, and self-recruitment from the backtracking simulations. The regions are the Detroit River Mouth (region A), Western Shoreline (region B), Midlake Reefs (region C), and Bass Islands (region D).

| $M_C$ | $B_{CA}$ | $B_{CB}$ | $B_{CC}$ | $B_{CD}$ | $R_{CA}$ | $R_{CB}$ | $R_{CC}$ | $R_{CD}$ | SR_$B_C$ |
|---|---|---|---|---|---|---|---|---|---|
| 19460 | 17 | 94 | 3778 | 2 | $8.7\times10^{-4}$ | 0.0048 | 0.194 | 0.0001 | 0.971 |
| 77840 | 75 | 337 | 15300 | 8 | $9.6\times10^{-4}$ | 0.0043 | 0.196 | 0.0001 | 0.973 |
| 155680 | 137 | 685 | 30659 | 22 | $8.8\times10^{-4}$ | 0.0044 | 0.197 | 0.0001 | 0.973 |
| $M_D$ | $B_{DA}$ | $B_{DB}$ | $B_{DC}$ | $B_{DD}$ | $R_{DA}$ | $R_{DB}$ | $R_{DC}$ | $R_{DD}$ | SR_$B_D$ |
| 18550 | 0 | 0 | 1911 | 407 | 0 | 0 | 0.103 | 0.022 | 0.176 |
| 74200 | 0 | 0 | 7628 | 1718 | 0 | 0 | 0.102 | 0.023 | 0.184 |
| 148400 | 1 | 0 | 15302 | 3430 | $6.7\times10^{-6}$ | 0 | 0.103 | 0.023 | 0.183 |

**Table 3.** The number of larvae and larval density (No. per cell) produced in the four regions. The regions are the Detroit River Mouth (region A), Western Shoreline (region B), Midlake Reefs (region C), and Bass Islands (region D).

| $M_C$ | $N'_{A\_C}$ | $N'_{B\_C}$ | $N'_{C\_C}$ | $N'_{D\_C}$ | $d'_{A\_C}$ | $d'_{B\_C}$ | $d'_{C\_C}$ | $d'_{D\_C}$ |
|---|---|---|---|---|---|---|---|---|
| 19460 | 550 | 3133 | 28406 | 10000 | 22.9 | 62.7 | 51.1 | 94.3 |
| 77840 | 2427 | 11233 | 115038 | 40000 | 101.1 | 224.7 | 206.9 | 377.4 |
| 155680 | 4434 | 22833 | 230519 | 110000 | 184.8 | 456.7 | 414.6 | 1037.7 |
| $M_D$ | $N'_{A\_D}$ | $N'_{B\_D}$ | $N'_{C\_D}$ | $N'_{D\_D}$ | $d'_{A\_D}$ | $d'_{B\_D}$ | $d'_{C\_D}$ | $d'_{D\_D}$ |
| 18550 | 0 | 0 | 44134 | 16545 | 0 | 0 | 79.4 | 156.1 |
| 74200 | 0 | 0 | 176166 | 69837 | 0 | 0 | 316.8 | 658.8 |
| 148400 | 126 | 0 | 353395 | 139431 | 5.3 | 0 | 635.6 | 1315.4 |

## 5 Discussion

We have shown, using both theoretical arguments and numerical data, that self-recruitment (SR) cannot be unambiguously computed using forward Lagrangian particle tracking models. In contrast, backward Lagrangian particle tracking models have demonstrated to be straightforward and effective in calculating SR.

SR depends on the larval production at each source location (Eq. 3), as noted by Lett et al. (2015). This dependence suggests that using forward tracking models to compute SR may be invalid if any strategy for larval particle release, such as releasing a random number, an equal number, or a number proportional to the area of the location, is employed, rather than releasing the realistic larval production from each source location. Our numerical data confirmed that variations in the release of larval particles from any source location can lead to different values of SR. This is in addition to the likelihood that there are unknown source locations contributing unexpected recruitment that are not accounted for in the simulation.





Similarly, researchers do not need to measure the larval production and dispersal rates of every potential source location to
estimate the SR at a given location. Instead, they can easily obtain the SR by estimating the number of local juveniles from
the total sampled juveniles at a given location, based on DNA relationships (D'aloia et al., 2013; Almany et al., 2017).

322        This shows that SR is independent of the number of larval recruits at the location of interest. This independence

makes it effective to compute SR using backward tracking models by releasing a random number of larval particles from the
location, as our numerical data demonstrated that SR had negligible variations with the number of larval particles released
from the settlement location. Despite the increasing usage of backtracking models to estimate larval hatching/spawning
locations and to study larval recruitment, few studies have used backtracking models to compute SR (Torrado et al., 2021).
Considering the limitations of backtracking models, for example that they are diffusive backward-in-time rather than being
convergent, comparisons with results from parentage analysis should be undertaken to further verify the validity of SR when
computed using backtracking models.

330        Local retention (LR) is typically more challenging to evaluate empirically, compared to SR, as sampling the

eggs/larvae that successfully grow into juveniles is more difficult than sampling recruits/juveniles (Lett et al., 2015). While
parentage analysis can identify the source of sampled juveniles, accurately accounting for the total number of juveniles
originating from a given source remains a challenge, as some juveniles are inevitably transported to unknown locations and
may be missed. For example, Almany et al. (2017) sampled adult and juvenile *Amphiprion percula* and *Chaetodon*
*vagabundus* from eight different locations in Papua New Guinea and assigned juveniles to their parents according to DNA
relationships. The location of their parents served as the source location of the juveniles, allowing the researchers to
determine the number of juveniles produced from each source location. However, the total number of larvae produced
(including those lost to mortality) remained unknown. The difficulty in sampling newly hatched larvae, i.e., measuring $N_i'$, is
likely why it is common to apply different larval particle release strategies from each source location in forward tracking
simulations.

341        Knowing the number of recruits $M_i'$, or the larval production $N_i'$ at one location, can allow us to estimate the number

of recruits and larval production at all other locations from Eqs. (11) and (12) using forward and backward tracking
simulations. An approach to estimate $N_i'$ was proposed (Eq. 10), based on the number of recruits at a settlement location $M_j'$,
the settlement rate $R_{ji}$ and the dispersal rate $D_{ij}$. From Eqs. (10) and (11), the $N_i'$ values can be computed based on the
recruits at different settlement locations and should be consistent. For example, $N_i'$ computed from the recruits at regions C
and D should be equal, such that $N_{i\_C}' = N_{i\_D}'$. However, large differences between $N_{i\_C}'$ and $N_{i\_D}'$ were modeled (Table 3).
When $D_{ij}$ or $R_{ji}$ approaches zero, the computation of $N_i'$ is undefined (section 2.3), which partly explains the difference.
Moreover, random numbers of recruits $M_j$ were released from regions C and D, which also caused differences between $N_{i\_C}'$
and $N_{i\_D}'$. These differences can be reconciled when $D_{ij}$ and $R_{ji}$ are non-zero by adjusting $M_D$ based on $M_C$; $N_{i\_C}'$ would then
equal $N_{i\_D}'$. For example, if $M_C = 19460$ particles released and backtracked from region C, the realistic number of recruits at
region C is $M_C' = M_C \cdot (R_{CA} + R_{CB} + R_{CC} + R_{CD}) = 3887$, the realistic recruits at region D, $M_D' = 1514$ would then be



estimated based on Eq. (12). The particles released and backtracked from region D should, therefore, be $M_D =$
$M'_D / R_{DA} + R_{DB} + R_{DC} + R_{DD} = 12112$ instead of 18550 in Table 3. The $N'_C$ value based on the recruits at region C, $N'_{C\_C}$,
would be equal to 28820 (from Eq. 11) and $N'_C$ estimated from the recruits to region D, $N'_{C\_D}$, would equal to 28811. The
negligible difference between 28820 and 28811 indicates the correctness of Eqs. (11) and (12).

## 6 Conclusions

Our findings show that self-recruitment (SR) is dependent on larval production at each potential source location that may
transport larvae to the location of interest. From this, we show theoretically and confirm using Lake Whitefish simulations,
that SR may not be computed unambiguously in forward tracking models without first identifying all the potential source
locations and their respective larval production. The latter becomes particularly evident given that four different strategies
for releasing larval particles from each source location have been employed. In contrast, in parentage analysis studies, it is
typically not necessary to measure the larval production and dispersal rates of every potential source location to estimate SR
at a given settlement location. Instead, by directly identifying the proportion of locally originating juveniles among the
sampled juveniles at a given location based on DNA relationships, SR can be determined more efficiently and accurately.
Similarly, releasing larval particles at the settlement location and tracking them backward in time offers a straightforward
approach to computing SR. Our findings demonstrated that SR is independent of the number of larval recruits at the
settlement location, making it viable to release a random number of larval particles. SR can thus be easily obtained as the
fraction of larval particles that settle locally, saving considerable effort and resources that would otherwise be spent
identifying all potential sources and their larval output. Furthermore, we proposed an approach to estimate larval production
at each source location by leveraging the connectivity between source and settlement locations, computed through
combining forward and backward tracking models. When run in isolation, backtracking models are only able to compute SR
(or theoretical SR) and the settlement rate, and forward tracking models are only able to compute LR (or theoretical LR) and
the dispersal rate. Whereas using a combination of both models allows for the calculation of not only SR and LR, but also
the larval production at each source location and the number of recruits at settlement locations. The ability to accurately
compute these metrics will significantly improve understanding of population connectivity. The findings were validated
using numerical data for the Lake Whitefish freshwater species but are also appliable to marine species with a pelagic larval
phase.





# Appendix A

**Table A1.** Release regions, times and durations for forward and backward tracking simulations.

|  | Release regions | Release times | Tracking periods (day) | Number of cells |
|---|---|---|---|---|
| Forward | Detroit River Mouth (Region A) | 21 March to 8 May | 12 | 24 |
|  | Western Shoreline (Region B) | 21 March to 8 May | 12 | 50 |
|  | Midlake Reefs (Region C) | 21 March to 8 May | 12 | 556 |
|  | Bass Islands (Region D) | 21 March to 8 May | 12 | 106 |
| Backward | Midlake Reefs (Region C) | 2 April to 20 May | 12 | 556 |
|  | Bass Islands (Region D) | 2 April to 20 May | 12 | 106 |

**Table A2.** The description of the notations in this research.

| Notation | Description |
|---|---|
| $B_{ij}$ | Number of particles recruited to site $j$ that were released from site $i$ in backtracking |
| $d'_i$ | Number of larvae per cell (500 m × 500 m) produced from site $i$ |
| $d'_{i\_j}$ | Number of larvae per cell (500 m × 500 m) produced from site $i$ estimated based on the recruits to site $j$ |
| $D_{ij}$ | Proportion of larvae released from site $i$ that recruit into the juvenile population at site $j$ |
| $F_{ij}$ | Number of particles recruited to site $j$ that were released from site $i$ in forward tracking |
| $LR_i$ | Ratio of local larval recruitment at site $i$ to the number of larvae released locally, that settled in suitable nursery sites |
| $M_i$ | Number of particles released from site $i$ in the backtracking simulations |
| $M'_i$ | Number of larval recruits to site $i$ |
| $N_i$ | Number of particles released from site $i$ in forward tracking simulations |
| $N'_i$ | Number of larvae produced at site $i$ |
| $N'_{i\_j}$ | Number of larvae produced from site $i$ estimated from the recruits to site $j$ |
| $N'_{ji}$ | Number of recruits at site $i$ that were released from site $j$ |
| $R_{ij}$ | Proportion of particles released from patch $i$ that settled in patch $j$ in the backtracking simulations |
| $SR_i$ | Ratio of local larval recruitment at site $i$ to all recruitment at site $i$ |
| $TLR_i$ | Ratio of local larval recruitment to site $i$ to local larvae released |





| TSR$_i$ | Ratio of particles that settled at site $i$ to all the particles released from site $i$ in backtracking simulations |
|---|---|

383

*Code and data availability*. The AEM3D executable was used as a black-box hydrodynamic transport code. The AEM3D source code was not modified in this application but is available with permission from HydroNumerics. The model setup for AEM3D are available at https://doi.org/10.5281/zenodo.14749408. The forward and backward particle tracking models were performed in Matlab. Their code and simulated date are all available at https://doi.org/10.5281/zenodo.14789098. The velocity output from AEM3D is also presented at https://doi.org/10.5281/zenodo.14789098.

*Author contributions*. WS conceived the main study design, developed the theory, performed the simulations and analyses. WS wrote the first draft of manuscript and LB and JDA revised the draft significantly. LB and SLS co-supervised the research and provided resources. JDA, LB, SLS and YMZ acquired research funding. All authors contributed to the project conceptualization, and editing and revising the manuscript.

*Competing interests.* The contact author has declared that none of the authors has any competing interests.

*Disclaimer*. Publisher's note: Copernicus Publications remains neutral with regard to jurisdictional claims in published maps and institutional affiliations.

*Financial support.* This work was supported by the Ontario Ministry of Natural Resources and Forestry, the Canada Ontario Agreement program and an NSERC Alliance Grant program to Josef D. Ackerman, Leon Boegman, and Shiliang Shan.

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
