# Peer review of "Computation of Self-recruitment in Fish Larvae using Forward- and"

_Geoscientific Model Development, 2024_

## Author Comment (AC2)

**#Comments from Romain Chaput**

I have carefully reviewed the manuscript by Shi et al., which presents a valuable theoretical and modelling framework for estimating self-recruitment (SR) in larvae by using both forward and backward particle tracking in a biophysical model. The approach is based on a Lagrangian model. This methodology is especially used in the context of population connectivity studies, where the accurate estimation of SR is important for ecological research, conservation planning, and ecosystem management.

The manuscript is clearly written, well-structured, and provides a strong demonstration of the methodology through a case study in Lake Erie. Notably, the authors show how this approach can be linked with genetic studies to validate or complement connectivity estimates, which significantly broadens the applicability of their method.

The study is timely and offers a potentially general tool for assessing self-recruitment, larval dispersal and connectivity in aquatic systems. Overall, I support the publication of this manuscript after minor revisions. My main concern relates to the number of particles used in the simulations and how this might influence the robustness of SR estimates.

Response: We sincerely thank Dr Chaput for the carefully reading, revision and recognition. The comments have been addressed in our point-by-point responses below.

**Minor Comments:**

**1. The manuscript would benefit from a more detailed discussion on whether the number of released particles is sufficient to saturate the system, especially given the stochasticity introduced in the simulations. A paragraph in the *Discussion* section addressing how particle number affects SR estimates, and the potential biases introduced by under-sampling, would strengthen the paper. Ideally, some justification or sensitivity analysis could be added or referenced.**

Response: We thank the reviewer for the insightful comment. In response, we have conducted a sensitivity analysis and added a paragraph in the Results Section. Our results on local retention and self-recruitment are reliable when more than 40000 particles are released in both forward and backward tracking simulations. However, releasing >40,000 particles in forward-tracking simulations does not inherently eliminate the self-recruitment (SR) biases caused by under-sampling. As shown in Eq. (3), such biases can only be avoided if realistic larval production is released from all potential source locations.

It is known that the tracking results could be sensitive to the number of particles released (Béguer-Pon et al., 2016). When too few particles are released, the

results fluctuate (Figure 3). To avoid these errors and achieve statistically stable results, a sensitivity analysis was conducted on the number of particles released. The results show that the estimated dispersal rate, local retention (Eq. 2) and self-recruitment are largely invariant when the number of particles released exceeds 40000 in both forward and backward tracking simulations (Figure 3).

[Figure]

Figure 3. A sensitivity analysis on the number of particles released. (a) In forward simulations, the dispersal rate $D_{ij}$ becomes statistically stable when more than 40000 particles are released. (b) In backtracking simulations, the self-recruitment $SR_i$ becomes statistically stable when more than 40000 particles are released.

**2. It would be valuable to include a short discussion on how larval mortality during dispersal and settlement could influence SR estimates. Furthermore, if the model outputs or the analytical framework could allow for the inference of mortality (e.g., when both larval production and SR are known), this should be briefly discussed.**

Response: We thank the reviewer for this comment, it's really a good point. We have written a description in the Discussion Section as the follows:

Local retention (LR) is typically more challenging to evaluate empirically, compared to SR (Lett et al., 2015). SR can be estimated by sampling recruits/juveniles at a location and determining the proportion of recruits that originate locally using DNA relationships. However, estimating LR requires sampling all of the eggs/larvae produced at a location that successfully grow into juveniles and disperse to many other locations, yet some juveniles may be transported to unknown locations and be missed. Theoretical local retention (TLR) is thus more challenging to evaluate, as it requires knowledge of the total number of larvae produced ($N_i'$), including those lost to mortality. For example, Almany et al. (2017) sampled adult and juvenile *Amphiprion percula* and *Chaetodon vagabundus* from eight different locations in Papua New Guinea and assigned juveniles to their parents according to DNA

relationships. The location of the parents served as the source location of the juveniles, allowing the researchers to determine the number of juveniles produced from each source location. However, it remained unknown whether all larvae were transported exclusively to these eight locations, as well as the total number of larvae produced. The difficulty in sampling newly hatched larvae, i.e., measuring $N_i'$, is likely why it is common to apply different larval particle release strategies (e.g., releasing a random, constant, or number of particles proportional to the area) in forward tracking simulations. As such, larval mortality (Mor) can affect the estimation of LR (Eq. 16). Specifically, increasing mortality reduces $\sum_{j=1}^{n} D_{ij}$ (the denominator of LR), while its impact on the numerator $D_{ii}$ remains unknown, causing the exact mechanisms by which mortality influences LR to be poorly understood. SR depends on either the recruitment rate (Eq. 5) or settlement rate (Eq. 7), and whether larval mortality affects these rates requires further research.

$$\text{Mor} = \frac{N_i' - \sum_{j=1}^{n} D_{ij} N_i'}{N_i'} = 1 - \sum_{j=1}^{n} D_{ij} = 1 - \frac{TLR_i}{LR_i} , \tag{16}$$

**Specific Comments:**

**1. Lines 15–17: The sentence beginning by "However, various strategies have been employed…" is somewhat confusing. Consider rephrasing for clarity.**

Response: We have revised the sentence as:
However, various strategies have been employed for releasing larval particles, including releasing a random number, a constant number, or a number particles proportional to the location area or the larval production. The lack of a consistent approach leads to ambiguous results.

**2. Line 231: The number of larvae released should be explicitly stated in the main text, along with a brief rationale for varying numbers across different release locations.**

Response: We have added the sentences in Section 3.3 as:
To study the impacts, on SR, of varying the number of particles released from a source location, three different numbers of particles were released from each region (Tables 1 and 2). For example, in forward tracking, 16800, 84000, 151200 particles were released in region A, and 17500, 70000, 140000 particles were released in region B. Each release region was divided into 500 m × 500 m AEM3D grid cells, and particles were released at the centers of the cells. Since the cell counts vary across regions (Table A1), for example there are 24 cells in region A while 50 cells in region B, we adjusted the number of particles released in each cell to standardize the total number of particles released per region. For example, we released 100, 500, 900 particles in each cell in region A (100 particles/cell/time × 24 cells × 7 times = 16800

particles), and 50, 200, 400 particles in each cell in region B (50 particles/cell/time $\times$ 50 cells $\times$ 7 times = 17500 particles).

**3. Line 232: Mention that the tracking duration corresponds to the pelagic larval duration (PLD) of the target species.**

Response: We have added a sentence as:
this tracking duration corresponds to the pelagic larval duration of Lake Whitefish.

**4. Line 235: Please clarify why forward tracking is conducted from four regions while backtracking is performed from only two. Does this reflect known settlement areas or observed recruitment patterns in Lake Erie?**

Response: We have clarified this in Section 3.3 as:
In our preliminary tests, backtracking simulations from regions A and B showed negligible settlement in any of the four regions (A, B, C and D), we thus restricted particle release to regions C and D in subsequent backtracking simulations for the sake of computational efficiency.

**5. Table 1: Define LR_FA clearly in the caption. Also, include a brief explanation of whether the particle release numbers are sufficient for system saturation.**

Response: We have added the description of $\text{LR\_F}_i$ in the caption. We have conducted a sensitivity analysis and added a paragraph in the Results Section as presented above.

**6. Table 2: Add the term SR$_{ij}$ to the caption.**

Response: We have added the description of $\text{SR\_B}_i$ in the caption.

**7. Table 3: The methodology used to derive the larval numbers in this table is not immediately clear. Are these values inferred from combined forward and backward simulations? Please clarify this in the caption with a brief methodological summary.**

Response: We have rewritten the caption of Table 3 as: The number of larvae produced in the four regions was computed based on the backtracking simulations from region C ($N'_{i\_C}$) and from region D ($N'_{i\_D}$) and the corresponding larval density (number per cell) $d'_{i\_C}$ and $d'_{i\_D}$.

A detailed description of $N'_{i\_C}$ and $d'_{i\_C}$ are given in the last paragraph of Section 3.4.

**8. Line 339: The phrase "different larval particle release strategies" needs clarification. Does this refer to spatial distribution, timing, number of particles, or something else?**

Response: We have clarified it as:

The difficulty in sampling newly hatched larvae, i.e., measuring $N_i'$, is likely why it is common to apply different larval particle release strategies (e.g., releasing a random, constant, or number of particles proportional to the area) in forward tracking simulations.

**This is a strong and valuable contribution, and with the suggested clarifications and additions, I believe the manuscript will be of high interest to the modelling and marine connectivity communities.**

Response: We sincerely thank the reviewer for the time and effort in reviewing our submission and these kind words.

**Reference**

Béguer-Pon, M., Shan, S., Thompson, K. R., Castonguay, M., Sheng, J., and Dodson, J. J.: Exploring the role of the physical marine environment in silver eel migrations using a biophysical particle tracking model, ICES Journal of Marine Science, 73, 57-74, 10.1093/icesjms/fsv169, 2016.

Lett, C., Nguyen-Huu, T., Cuif, M., Saenz-Agudelo, P., and Kaplan, D. M.: Linking local retention, self-recruitment, and persistence in marine metapopulations, Ecology, 96, 2236-2244, 10.1890/14-1305.1, 2015.

---

## Author Comment (AC3)

**Comments from Reviewer #2**
**I got very excited by this work reporting the possibility to simulate self-recruitment (SR) without the need of information on larval production, thereby solving a long-time reported issue for comparing models and data of larval connectivity: specifically, biophysical models are appropriate tools to simulate local retention (LR) whereas field studies estimate SR.**

**Response:** We sincerely thank the Reviewer for reading and commenting on our work. The comments have been addressed in our point-by-point responses below.

We agree that local retention (LR) is typically more challenging to evaluate empirically, compared to self-recruitment (SR) (Lett et al., 2015). However, note that Almany et al. (2017) estimated LR using field-based parentage analysis.

**1. Unfortunately, I believe that the proposed method implicitly assumes spatially homogeneous larval production and is therefore no more appropriate for simulating SR than the usual method. The assumption of homogeneous larval production is here implicit, because of the use of backward-in-time tracking, as opposed to being explicit when using forward-in-time tracking.**

**Response:** The reviewer does not explain why they believe homogeneous larval production would occur in an implicit tracking scheme or with an implicit assumption, so it is, therefore, difficult to respond to this concern.

Specifically, it is unclear whether the reviewer is referring to an implicit numerical scheme, or an implicit assumption. An implicit numerical scheme computes the state of a system using information from both the present and the next time step. An explicit scheme computes the state of a system using only information from the present time step. Our forward/backward tracking scheme is an explicit scheme (see Shi et al 2024; Eq. 1 & 2), using only present time step velocities computed with an independent hydrodynamic model, saved as snapshots every 12 time steps. The only difference is that we reverse the sign of the velocity vector. Therefore, our model is explicit, not implicit.

An implicit assumption means something is understood or implied but not directly stated. Backtracking models, ours and in general, do not assume spatially homogeneous larval production. On the contrary, the source location and larval production are quantities that backtracking models aim to simulate, which was presented in Eq. 11 in this paper.

From a mathematical standpoint, there is no implicit requirement that the probability density distribution, $P(t_i, X; t_f, Y)$, representing the distribution of particles (larvae, in this context), be spatially homogeneous. The temporal evolution of $P$ is governed by the Kolmogorov forward and backward equations, first introduced by Kolmogorov

(1931; English translation in Kolmogorov, 1992). These equations mathematically describe how particle distributions evolve under stochastic processes involving advection and diffusion. Specifically, the backward-in-time tracking methodology involves solving the Kolmogorov backward equation from a known final distribution at time $t_f$ (larval settlement time) backward toward the unknown initial distribution at time $t_i$ (larval production time). The transport processes (advection and diffusion) employed in these equations are independently determined from hydrodynamic model outputs. Therefore, the backward-in-time tracking approach does not inherently assume that the initial probability density distribution at $t_i$ must be spatially homogeneous (i.e., a uniform distribution). Instead, the primary goal of backward tracking is explicitly to reconstruct the initial probability density distribution at time $t_i$ based on the known distribution at time $t_f$.

In this research, we released larval particles at known nursery sites and track them backward in time, using an explicit scheme, to determine how many particles may have been advected from the source locations using independently evaluated hydrodynamics. There are no assumptions on how many were produced at these locations.

If backtracking models assumed that each potential hatching location produced a consistent number of larvae, then the modeled density of larval particles that settle at each location should be consistent, which means backtracking models are meaningless since the modeled results are known to be consistent. However, as presented by many studies that used backtracking models to estimate spawning/hatching grounds of fish species, larval particles settled at each location with different densities (e.g., Figure 3b in Shi et al., 2024; Rowe et al., 2022; Torrado et al., 2021; Gargano et al., 2022). This is like larval settlement not being implicitly assumed to be spatially homogeneous in forward tracking models.

In this research, we estimated SR using forward tracking simulations and assuming larval production was consistent at four source regions. The SR at regions C and D was estimated as 0.68 and 0.32, respectively. If backtracking simulations used an implicit scheme that assumed spatially homogeneous larval production, the SR computed using backtracking model should equal to the SR above. However, using the backtracking model, SR at regions C and was estimated as 0.97 and 0.18, respectively. This demonstrates that backtracking simulations have not implicitly assumed spatially homogeneous larval production.

**Indeed, imagine that a given zone is particularly suitable for larval production in the reality. In the model there is no reason why more particles would be advected back to that zone "hydrodynamically" . To represent this "biological" reality of enhanced larval production the number of particles advected back to the given zone should be weighted by larval production from that zone in order to**

**calculate SR correctly. If not, then homogeneous larval production is implicitly assumed.**

**Response:** We disagree with the reviewer. The hydrodynamics change spatially and temporally in the lake on a 5 min time step. Based on observations, we released particles at known first-feeding times in known nursery locations. This represents the biological reality of the larvae. It is then up to the hydrodynamic model to track them backward in time to potential spawning locations. The hydrodynamics could also transport some or all the particles backward in time to unsuitable spawning locations, which would result in no larval production in those locations. In this study, we do not simulate biology, but rather how the flow field regulates where the particles are advected backward to, how many particles are advected there and if they are known spawning sites.

Note that the number of larvae being transported from a source zone to a settlement zone is not controlled by the larval production of the source zone, it depends on both larval production and dispersal rate. For example, source zones A, B, and C produce 200, 150, 100 larvae, respectively, but if the dispersal rate from these source zones to a settlement zone Z is 0.1, 0.6, 0.9, then the number of larvae transported from source zones to the settlement zone Z are 20, 90, 90. When performing backtracking simulations and releasing 1000 particles from zone Z, 1000×20/(20+90+90)=100 particles will be transported to zone A. But in addition, 450 and 450 particles will be transported to zones B and C. Source zone A has a highest larval production but the lowest number of particles advected back to it in backtracking simulations.

**2. It is true that the quantity simulated using back-tracking will not depend on the number of particles released in a settlement area, as long as this number is high enough to obtain a statistically meaningful value. However that simulated quantity will be comparable to SR as assessed in the field only if larval production is spatially homogeneous.**

**Response:** We disagree with the reviewer. In forward tracking simulations, SR does depend on the larval production and dispersal rates, as presented in Eq. 3. However, in backtracking simulations, SR depends only on the settlement rate, with no relationship to larval production, with larval settlement as presented in Eq. 7. The settlement rate is terminology specific to backtracking simulations, defined as the ratio of particles that settle locally at a location divided by the total number of particles released from that location. This is similar to forward tracking simulations, where LR only depends on the dispersal rate, while in backtracking simulations, LR depends on both larval settlement and the settlement rate. However, we cannot deduce that spatially homogeneous larval settlement has been implicitly assumed in forward tracking simulations. The factor that regulates both the dispersal rate and settlement rate is the hydrodynamics, which controls the movement trajectories of larval particles.

Moreover, the hydrodynamic model (AEM3D) has been shown to accurately simulate the flow field in Lake Erie. The AEM3D model, and its non-parallel predecessor the Estuary and Lake Computer Model (ELCOM), have been applied to hindcast the thermal structure (León et al., 2005), internal wave dynamics (Valipour et al., 2015), surface wave / sediment transport (Lin et al., 2021), nutrient and chlorophyll-a distributions (Leon et al., 2011), seasonal succession of phytoplankton groups (Wang et al., 2024); and to forecast storm surge and upwelling/downwelling events (Lin et al., 2022). In our prior work, we applied the model, to Lake Erie, to backtrack the present Lake Whitefish larval observations and determine hatching locations (Shi et al., 2024). As we have discussed, the biggest drawback with using hydrodynamic models for backtracking, is that they are diffusive backward-in-time, rather than being convergent. Therefore, comparisons with results from parentage analysis should be undertaken to further verify the validity of SR when computed using backtracking models.

**3. In conclusion I sadly do not see how the proposed method based on back-tracking is an improvement from the one based on forward-tracking. Back-tracking is an approach that may be more efficient to use computationally than forward-tracking when focusing on particles origin rather than destination but both approaches should give the same results.**

**Response:** In the present research, we show that, using both theoretical arguments and numerical simulations of Lake Whitefish (*Coregonus clupeaformis*) larval observations in Lake Erie, forward tracking cannot compute SR accurately unless realistic larval production is released from all potential source locations. However, backtracking simulations can easily compute SR with no knowledge of larval production. This is shown theoretically in the comparison of Eq. (3) and Eq. (7).

We employed various strategies for releasing larval particles in the forward tracking simulations, including releasing a random number, a constant number, or a number of particles proportional to the location area or the larval production. In all cases, the numerical simulations of Lake Whitefish (*Coregonus clupeaformis*) in forward tracking cannot compute SR unless realistic larval production values were released from all potential source locations. While in contrast, tracking larval particles backward from the settlement location was shown to be a straightforward approach for computing SR. Similarly, backtracking models cannot compute LR unless a realistic number of larval recruits are released from all settlement locations.

**4. lines 40-48 I found this paragraph confusing as it looks like the authors decided to define local retention (LR) and theoretical local retention (TLR) oppositely to what was previously used (Hogan et al. 2012, Burgess et al. 2014, Lett et al. 2015).**

**Response:** Yes, Hogan et al. (2012) and Lett et al. (2015) proposed a different measure to assess local retention.

- Botsford et al. (2009) defined LR as: $\dfrac{\text{locally produced settlement}}{\text{total number of larvae locally released}}$

- Hogan et al. (2012) defined LR as: $\dfrac{\text{locally produced settlement}}{\text{total number of settlers locally released}}$

which was termed relative local retention (RLR) by Lett et al. (2015).

- More recently, Almany et al. (2017) defined LR as:
$$\dfrac{\text{number of larvae that left reef X and survived and settled on reef X}}{\text{number of larvae that left reef X and survived and settled somewhere else}}$$

which corresponds to the RLR, defined by Lett et al. (2015).

Consequently, there are two definitions of local retention in the literature. The difference between these two terms lies in the denominator. Hogan and Almany's LR, includes only successfully settled larvae, Botsford and Lett's LR, encompasses both successful and unsuccessful settlers (those settling in unsuitable nursery locations). In field studies, sampled fish are typically survivors. To ensure comparability with empirical results, we adopt Hogan's LR, as local retention, while Botsford's LR, is termed "theoretical" local retention (TLR). TLR provides a minimum value of LR as stated by Shi et al. (2024), as the denominator contains both successfully and unsuccessfully settlers.

**5. 49: Parentage analysis and/or larval tagging is widely used to estimate SR, not the other quantities, to my knowledge.**

Response: We agree. Almany et al. (2017) used parentage analysis to estimate local retention. We have rewritten this sentence as:

Parentage analysis and/or larval tagging have been widely used to estimate LR and SR (Almany et al., 2017; Jones et al., 1999; Pinsky et al., 2012; D'aloia et al., 2013; Lett et al., 2015; Planes et al., 2009).

**6. 69-72: True, but as explained above I believe this issue remains when using back-tracking.**

**Response:** As we explained above, we have not no backtracking models implicitly assumed spatially homogeneous larval production.

**7. 87-91: Same here.**

**Response:** As we explained above, we have not implicitly assumed spatially homogeneous larval production.

Response: As we explained above, no backtracking models implicitly assumed spatially homogeneous larval production.

**8. 103-4: It has already been shown that LR and TLR are independent on larval production (Burgess et al. 2014, Lett et al. 2015) and there is no need of simulations to prove this.**

Response: That LR and TLR are independent of larval production is not the conclusion of this research. We used the mathematical methods to derive LR, TLR and SR. The derived LR and TLR agree with previous research (Lett et al., 2015), which verifies the accuracy of our derived results, that SR depends on larval production but is independent on the number of recruits at a settlement location.

**9. Equ. (1) this quantity is defined as LR in Lett et al. (2015), why change?**

**Response:** As we explained above in comment #4, there are two definitions of local retention; the difference between these two definitions lies in the denominator. Hogan and Almany's LR, includes only successfully settled larvae, Botsford and Lett's LR, encompasses both successful and unsuccessful settlers (those settling in unsuitable nursery locations). In field studies, sampled fish are typically survivors. To ensure comparability with empirical results, we adopt Hogan's LR, as local retention, while Botsford's LR, is termed "theoretical" local retention (TLR). TLR provides a minimum value of LR as stated by Shi et al. (2024), as the denominator contains both successfully and unsuccessfully settlers.

**10. Equ. (2) this quantity is defined as RLR in   Lett et al. (2015), why change?**

**Response:** As indicated in response (9) we have chosen to pursue alternate terminology.

**11. 167-8 This is true, however as explained above that simulated quantity will be SR as assessed in the field only if larval production is spatially homogeneous.**

**Response:** The question about spatial homogeneity was addressed above in responses (1) to (3).

**291-2 I don't understand why SR values estimated from back-tracking and forward-tracking would be different. I don't agree that one is "correct" and the other "erroneous". They should be similar as both are based on the same assumption of homogeneous larval production, as explained above. There may be numerical reasons for the reported differences, or other technical reasons that could be explored.**

**Response:** This was addressed at length above in responses (1) to (3). That SR values estimated from backtracking and forward-tracking are different is the key finding of this research. We have shown this outcome both theoretically and numerically. Therefore, it is not for technical reasons that SR from forward tracking is different from backtracking.

When estimating SR using forward tracking models, SR varies with the number of larval particles released from each source location. Notably, the value of SR can be artificially manipulated to be any value from 0 to 1, by adjusting the quantity of larvae released. The correct SR from forward tracking simulations can only be obtained when realistic larval production is used. Consequently, SR from a backtracking model will only match that from forward tracking if realistic larval production numbers are used in the forward tracking simulations.

**References**

[revised manuscript text omitted]